# Splicing-aware scRNA-Seq resolution reveals execution-ready programs in effector Tregs

Daniil K. Lukyanov[1,2,3], Evgeniy S. Egorov[3,4], Valeriia V. Kriukova[5], Denis Syrko[2,3], Victor V. Kotliar[3], Kristin Ladell[6,7], David A. Price[6,7], Andre Franke[5], Dmitry M. Chudakov[1,2,3,8,9]*

**1** Center for Molecular and Cellular Biology, Moscow, Russia, **2** Institute of Translational Medicine, Pirogov Russian National Research Medical University, Moscow, Russia, **3** Genomics of Adaptive Immunity Department, Shemyakin and Ovchinnikov Institute of Bioorganic Chemistry, Moscow, Russia, **4** Faculty of Bioengineering and Bioinformatics, Lomonosov Moscow State University, Moscow, Russia, **5** Institute of Clinical Molecular Biology, Kiel University, Kiel, Germany, **6** Division of Infection and Immunity, Cardiff University School of Medicine, University Hospital of Wales, Cardiff, United Kingdom, **7** Systems Immunity Research Institute, Cardiff University School of Medicine, University Hospital of Wales, Cardiff, United Kingdom, **8** Central European Institute of Technology, Brno, Czech Republic, **9** Abu Dhabi Stem Cell Center, Al Muntazah, United Arab Emirates

* ChudakovDM@gmail.com

## Abstract

Single-cell RNA sequencing (scRNA-Seq) provides valuable insights into cell biology. However, current scRNA-Seq analytic approaches do not distinguish between spliced and unspliced mRNA at the level of dimensionality reduction. RNA velocity paradigm suggests that the presence of unspliced mRNA reflects transitional cell states, informative for studies of dynamic processes such as embryogenesis or tissue regeneration. Alternatively, stable cell subsets may also maintain translationally repressed spliced mRNA (e.g., in P-bodies) and/or unspliced mRNA reservoirs for prompt initiation of transcription-independent expression. Thus, functional cell subsets may differ not only in the current levels of actively produced mRNAs, but also in which mRNAs and in what forms are stored in the nucleus and cytoplasm. To enable splicing-aware analysis of scRNA-Seq data, we developed a method called SANSARA (Splicing-Aware scrNa-Seq AppRoAch). We employed SANSARA to characterize peripheral blood regulatory T cell ($T_{reg}$) subsets, revealing a complementary interplay between the FOXP3 and Helios master transcription factors and high levels of spliced *IL10RA*, *LGALS3*, *FCRL3*, *CD38*, *ITGAL*, and *LEF1* mRNAs in effector $T_{reg}$s. Among Th1 and cytotoxic CD4+ T cell subsets, SANSARA also revealed substantial splicing heterogeneity across subset-specific genes. SANSARA is straightforward to implement in current data analysis pipelines and opens new dimensions for scRNA-Seq-based discoveries.

**Data availability statement:** Raw scRNA-sSeq data can be accessed on NCBI Sequence Read Archive under the BioProject PRJNA995237 accession. Tables with saGEX and GEX values as well as our custom code pipeline are available at Github (https://github.com/EvgenEgorov/SANSARA).

**Funding:** This study was supported by RSF grant №25-75-30013, to DMC. DAP was supported by the PolyBio Research Foundation (Balvi B43). The funders had no role in study design, data collection and analysis, decision to publish, or preparation of the manuscript.

**Competing interests:** The authors have declared that no competing interests exists.

## Author summary

Single-cell transcriptomics classifies cells by the patterns of genes they express. Most methods, however, treat every RNA message in the same way, even though cells produce RNA in two stages: unspliced (nascent) and spliced (mature and ready to make protein). To provide additional resolution, we developed SANSARA, a splicing-aware analysis that uses this extra layer of information to sharpen how we read cellular states.

We applied SANSARA to human regulatory T cells ($T_{reg}$s) – immune cells that prevent harmful inflammation – which uncovered features that were missed by splicing-unaware analysis. SANSARA revealed unexpectedly complementary splicing behavior of genes encoding FOXP3 and Helios, the two major $T_{reg}$ transcription factors. Effector $T_{reg}$s were enriched for mature, translation-ready transcripts encoding key functionality, including MHC-II – antigen-presentation machinery, CD39 and CD38 – contributing to the generation of immunosuppressive adenosine, LFA-1 – stabilizes $T_{reg}$ interactions with dendritic cells, LEF1 – transcription factor that cooperates with FOXP3, and IL10RA – receptor that forms a feed-forward loop with IL-10, also produced by $T_{reg}$s.

This splicing-aware view provides a clearer picture of immune function and uncovers mechanisms that standard approaches often overlook. SANSARA transforms the interpretation of single-cell transcriptomics data and can be broadly applied to other cell types and diseases to deepen biological insight and guide target discovery.

## Introduction

RNA processing is an integral part of the implementation of genetic information [1,2]. Correspondingly, rational utilization of splicing information in scRNA-Seq data analysis could reveal multiple functional aspects of cell biology. However, quantitative analysis of splicing is rarely included in scRNA-Seq studies due to the difficulties inherent to the short-read sequencing technologies [3,4]. Coverage bias across genes and sequencing technologies, inability to detect all splicing junctions, insufficient sequencing depth, and high dropout rate prevent direct estimation of splicing by distinguishing spliced and unspliced molecules [3].

To date, splicing was studied in scRNA-Seq data in terms of transcriptional dynamics and cell-state transitions [5,6], and only in a *post hoc* manner – after conventional clustering and dimensionality reduction. However, splicing information has not been used as an independent criterion to distinguish between stable functional cell subsets, implemented as an input at the level of cell clustering.

At the same time, certain cell subsets may preferentially accumulate unspliced primary transcripts in the nucleus, serving as transcription-independent reservoirs for rapid production of mature mRNA and proteins [7,8], whereas a predominance

of spliced mRNA may be associated with effector cell states and/or mature mRNA reservoirs, such as P-bodies [9–12]. The same logic may be applicable to the non-coding RNA transcripts [13]. This means that one could consider the presence of certain spliced or unspliced RNA transcripts as a distinguishing feature for stable or relatively stable cell subsets, theoretically enabling the construction of splicing-aware scRNA-Seq data and the identification of corresponding functional cell clusters.

In this work, we describe SANSARA (Splicing-Aware scrNa-Seq AppRoAch), a method that produces splicing-adjusted gene expression matrix (saGEX) that accounts for the extent of splicing for each gene in each cell. The resulting saGEX is then subjected to a conventional clustering and dimensionality reduction pipeline to reconstruct a splicing-aware representation of the scRNA-Seq data.

We employ SANSARA to resolve the complexity of human peripheral blood helper T cells. This splicing-aware approach yields a deep structuring of the intrinsic heterogeneity of regulatory T cells ($T_{reg}$s) and the Th1/cytotoxic axis of helper T cells. We anticipate that SANSARA should have broad applications in single-cell transcriptomics beyond T cell biology, revealing a universe of distinctive and informative splicing-related features of tissue cell subsets.

## Results

### Splitting gene expression into spliced and unspliced values

Direct estimation of the proportion of spliced versus unspliced mRNA for each gene in scRNA-Seq data is confounded by the oligo-dT primers used to enrich for polyadenylated mRNA molecules, and the limited coverage and biases of currently-available information obtained via either 5'- or 3'- high-throughput transcriptomics [3]. We settled on the veloVI framework [14], which is based on the proportions of spliced and unspliced unique molecular identifiers (UMIs), where each UMI-labeled molecule containing a read mapping to an intronic region is counted as an unspliced molecule. These algorithms were initially developed for the determination of 'RNA velocity' [6], a parameter that reflects a transcriptomic snapshot of current mRNA turnover. Here we employed the veloVI-derived values to analyze cell heterogeneity using splicing-aware clustering and dimensionality reduction, in order to differentiate stable cell clusters characterized by distinct gene splicing features (Fig 1).

Initial velocyto-derived values [6] depend on individual gene features and cannot be employed for informative analysis. The downstream veloVI-derived values represent a much more accurate individual estimation of the extent of gene splicing based on the inferred gene-specific rates of transcription, splicing, and degradation [14]. We used the veloVI filtering steps and confidence scores to choose the subset of genes most suitable for splicing estimation. Next, we used veloVI-derived values to calculate the splicing-adjusted gene expression (saGEX) for each gene. saGEX is determined by multiplying the veloVI value by the total normalized expression of that gene in each cell (GEX) and assigned to either the spliced (negative splicing score values) or the unspliced (positive splicing score values) form of the gene in each cell (Fig 1).

The resulting saGEX cell-feature matrix simulates gene expression patterns conventionally used by dimensionality reduction methods, but split to discriminate spliced and unspliced gene forms. Finally, the saGEX data are analyzed using a standard Seurat pipeline with Seurat lognormalization, similar to conventional GEX analysis. This approach, named SANSARA, proved to provide natural and informative downstream analyses, as demonstrated in the following examples.

### Resolving splicing differences in scRNA-Seq landscape

To test SANSARA, we used datasets of sorted, effector-enriched CD4+ T cells from peripheral blood mononuclear cells (PBMC) of three donors, which were extensively characterized previously [15]. Notably, these CD4+ 5'-RACE 10x Genomics scRNA-Seq datasets were of high quality, with more than 5,000 median UMIs per cell, sequenced with relatively long reads (100 + 100 nt) and high coverage of 90,000 reads per cell. This may be crucial for performance of the veloVI and SANSARA algorithms. Spliced UMI without introns accounted for approximately 75% of counts per cell with no difference

PLOS Computational Biology

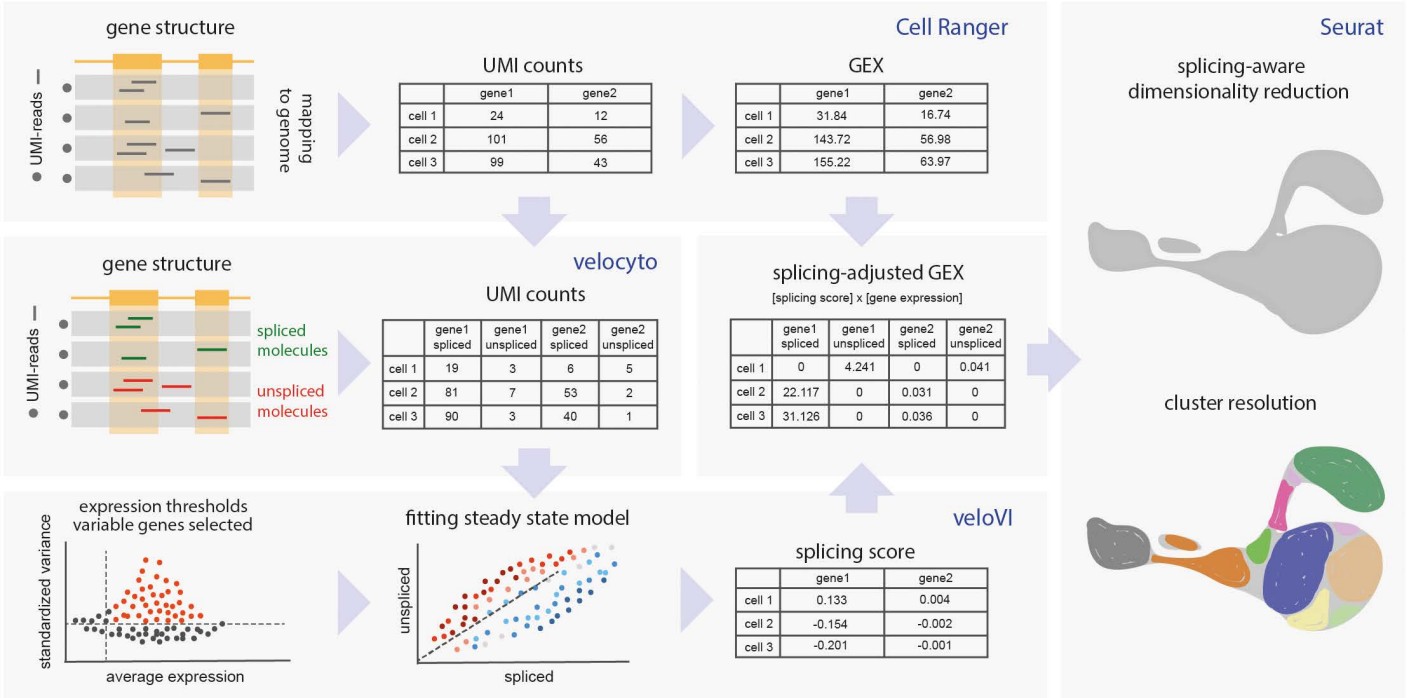

**Fig 1. SANSARA workflow.** After mapping scRNA-Seq data to the genome with Cell Ranger, spliced and unspliced UMI counts are differentiated using velocyto. Highly variable genes are selected based on log-normalized splicing-aware counts, and veloVI model is fitted to each gene. Genes for downstream analysis are chosen based on the quality of the fit. The product of original gene expression (GEX) and splicing score, termed splicing-adjusted GEX (saGEX) is then used for conventional dimensionality reduction and clustering analysis.

between clusters, and about 60% of counts per variable gene used in the downstream analysis, with higher variance (S1a-c Fig).

After obtaining saGEX values for ~1,500 genes from individual donors, we integrated them using the Harmony pipeline [16] to remove donor-specific batch effects (S2a and S2b Fig). Harmony algorithm was chosen because it operates on the level of low-dimensional PCA embedding and does not require raw counts with negative binomial assumption, as PCA uses scaled and centered data. The distribution and mean-variance relationship of non-zero saGEX expression values (as calculated by Seurat) were generally preserved relative to the conventional analysis, supporting the applicability of Harmony (S1d-g Fig).

The original splicing-unaware GEX datasets were analyzed separately using the same parameters for integration and dimensionality reduction. The general topology of the resulting splicing-aware UMAP data representation closely resembled that of the conventional splicing-unaware dataset, preserving the major subset composition (Fig 2a and 2b). Splicing-aware UMAP plots were characterized by higher clustering stability at different resolutions (S2c-e Fig).

We used several metrics to evaluate clustering performance of splicing-aware dimensionality reduction with the SANSARA approach. First, we calculated Silhouette scores on multiple resolutions and compared between splicing unaware and splicing aware datasets. For each resolution value, we computed the average silhouette score across all resulting clusters. This analysis showed that SANSARA outperforms conventional analysis on most resolutions (S3a Fig).

Next, we compared how faithfully a low-dimensional embedding (UMAP) preserves the neighborhood structure of a higher-dimensional reference space (Harmony-corrected PCA) across methods, as SANSARA values are different from

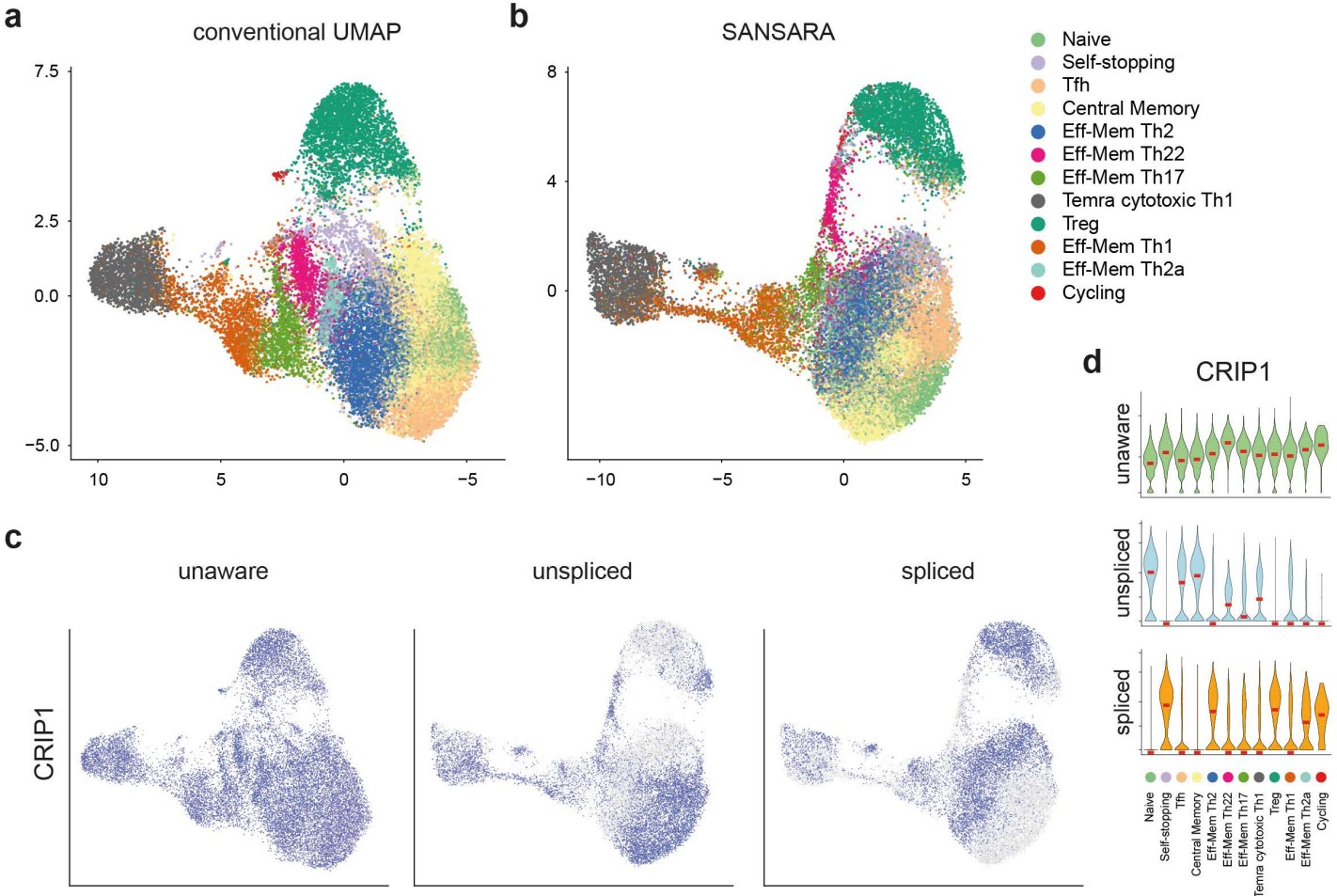

**Fig 2. SANSARA reveals splicing heterogeneity of CD4+ T cells. a,b.** Comparison of cluster annotation between the conventional splicing-unaware (a) and splicing-aware (b) UMAP plots. Annotated according to Ref. 11. c. UMAP plots of conventional GEX (left) versus saGEX (center, right) *CRIP1* expression. d. Violin plots of splicing-unaware (top) and -aware (middle, bottom) *CRIP1* expression across clusters.

conventional expression values. Trustworthiness is a metric that goes down if UMAP invents spurious neighbors (cells that weren't close in PCA space), and continuity assesses the loss of true neighbors in UMAP compared to PCA. Both scores are in (0, 1) range and are calculated at several k values (the neighborhood size). The results show that SANSARA analysis consistently preserves the PCA neighborhood structure at comparable level to the conventional splicing-unaware analysis (S3b Fig).

We also assessed the correspondence between clusters produced by splicing-unaware analysis and SANSARA at the same resolution and adjusted Rand Index (ARI), which measures the proportion of cell pairs that remain in the same clusters across methods while accounting for cluster sizes (S3c Fig). Both analyses showed that most of the clusters are nearly identical between the methods, with the exception of Naive, Tfh and CentMem, which are traditionally challenging to define.

Based on these results, we conclude that integration using single-cell transcriptome data with splicing taken into account is comparable to conventional scRNA-Seq data integration. Both approaches performed similarly, even though the splicing-aware dataset contains fundamentally different information.

Indeed, accounting for splicing painted a distinct picture of gene expression heterogeneity across subsets of helper T cells, with relatively uniform expression of many genes giving way to highly specific expression patterns based on splicing. Illustrative analysis of one such gene, *CRIP1*—encoding an intracellular zinc transport protein typically expressed in effector memory CD4+ T cells [17] is shown on Fig 2c and 2d. Other examples of relevant genes with heterogeneous splicing behavior include *FOS*, *ANXA1*, *TCF7*, *INPP4B*, and *MALAT1* (S4 and S5 Figs and S1 Table).

## SANSARA investigation of $T_{reg}$ subsets

We performed an analysis of the $T_{reg}$ subpopulation of CD4+ T cells [18,19] in order to assess what functionally relevant information could be unearthed with the use of splicing-aware scRNA-Seq analysis. At several resolutions, SANSARA consistently distinguished three major $T_{reg}$ clusters (Fig 3a-3d), corresponding to *naïve*, *activated*, and *effector* $T_{reg}$s, as classified in a recent deep scRNA-Seq investigation [20]. We have retained these cluster designations for consistency. Splicing-aware $T_{reg}$ clusters (Fig 3d) mapped similarly on the splicing-unaware UMAP (Fig 3c). Corresponding clusters could be also identified in splicing-unaware analysis (Fig 3a), and localized similarly yet not identically within the splicing-aware UMAP (Fig 3b).

Many of the revealed differences in expression of spliced versus unspliced transcripts were unexpected and informative and could thus meaningfully shape our understanding of the underlying functional state of different $T_{reg}$ subsets (Figs 3e-3h and S4-S6).

In particular, the gene encoding the $T_{reg}$ master transcription factor FOXP3 [21,22], was mostly expressed in the unspliced form in *naïve* $T_{reg}$s, presumably reflecting their readiness yet not involvement in active regulatory functions. In *activated* and *effector* $T_{reg}$s, *FOXP3* was mostly expressed in a spliced form. In contrast, another $T_{reg}$-characteristic transcription factor, *IKZF2* (Helios) was expressed in the unspliced form in *activated* and *effector* $T_{reg}$s, while *naïve* $T_{reg}$s predominantly contained spliced *IKZF2* mRNA (Fig 3e-3h). Previous data from mouse models have shown that the Helios transcription factor ensures $T_{reg}$ survival and lineage stability through activation of the IL-2Rα–STAT5 pathway and STAT5-dependent stabilization of *FOXP3* expression [23,24]. Our data indicate that the interplay between these two transcription factors may be more complex at the level of splicing regulation.

Activated and effector $T_{reg}$s were respectively characterized by expression of unspliced and spliced forms of *DUSP4* (dual-specificity phosphatase-4) (Figs 3f and S6), which encodes a protein that is involved with the regulation of STAT5 protein stability [25].

All CD4+ T cells expressed *IL10RA* according to conventional GEX analysis, but SANSARA revealed that the spliced form of *IL10RA* was almost exclusively observed in *activated* and *effector* $T_{reg}$s. Unspliced *IL10RA* expression was more prominent in *naïve* $T_{reg}$s and non-$T_{reg}$ CD4+ T cells (Fig 3f-3h). Expression of IL10RA on $T_{reg}$s is important for a feed-forward loop in which IL-10RA signaling reinforces IL-10 secretion by $T_{reg}$s, critical for proper control of Th17 subset activity [26,27].

We also observed a number of other cluster-specific patterns of splicing behavior. The spliced form of *LGALS3* (encoding galectin-3) was predominantly present in the *effector* $T_{reg}$ cluster, while the unspliced form was present in *naïve* and *activated* $T_{reg}$s (Figs 3f and S6). Galectin-3 has been shown to regulate $T_{reg}$ frequency and function in mouse models of *Leishmania major* infection [28] and autoimmune encephalomyelitis [29]. Reports have also shown that *LGALS3* expression is increased in human $T_{reg}$s through a transcriptional mechanism involving the ubiquitin D (*UBD*) gene, which is a downstream element of FOXP3 [30].

The *activated* $T_{reg}$ cluster was previously shown to express increased levels of *FCRL3* gene encoding Fc receptor-like protein 3 [20]. FCRL3 receptor stimulation of $T_{reg}$s has been shown to inhibit their suppressive function and induce IL-17, IL-26, and IFNγ production as well as expression of the Th17-defining transcription factor RORγt [31]. SANSARA revealed that spliced *FCRL3* is mostly expressed in effector $T_{reg}$s, potentially linking FCRL3 to self-restraint of effector $T_{reg}$ function (Fig 3f).

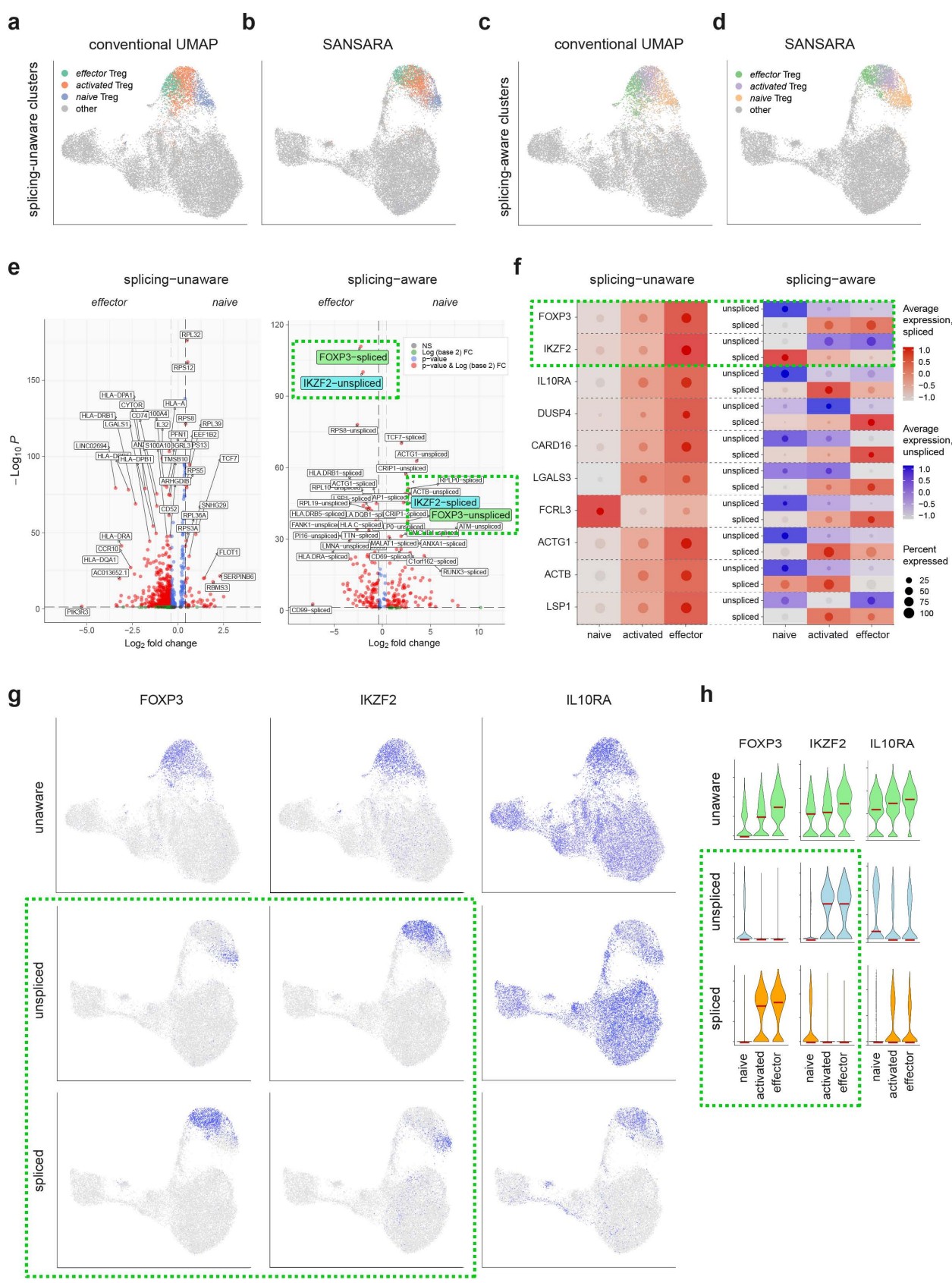

**Fig 3. Splicing-aware investigation of T$_{reg}$ heterogeneity.** a-d. Cross-positioning of splicing-unaware (a, b) and splicing-aware (c, d) naive, activated, and effector T$_{reg}$ clusters in splicing-unaware (a, c) and splicing-aware (b, d) datasets. Splicing-unaware *activated* T$_{reg}$ cluster (a,b) is a product of merging of the two corresponding clusters, see S1c Fig, resolution 2.0). e. Volcano plots of differentially-expressed genes between *naive* and *effector* T$_{reg}$ clusters from splicing-unaware (left) and -aware (right) datasets. f. Dot plot of standardized scaled expression of selected genes in three T$_{reg}$ clusters in splicing-unaware (left) and -aware (right) datasets. The diameter of the dot shows the proportion of cells expressing the gene. Background heatmap color corresponds to the color of the dot and reflects average expression of unaware or spliced (red) or unspliced (blue) gene forms. g. Splicing-unaware (top) and SANSARA (middle, bottom) UMAP plots showing expression of *FOXP3, IKZF2,* and *IL10RA*. h. Violin plots showing conventional splicing-unaware (top) and splicing-adjusted (middle, bottom) *FOXP3, IKZF2,* and *IL10RA* expression across three T$_{reg}$ clusters. Dashed green rectangles highlight expressions of spliced and unspliced *FOXP3* and *IKZF2*.

*Naïve* T$_{reg}$s preferentially expressed unspliced transcripts of the cytoskeleton-related protein genes *ACTG1* and *ACTB* [32], whereas expression of the spliced forms of these transcripts was more characteristic of *activated* T$_{reg}$s (Figs 3f, S4 and S6). The spliced form of the *LSP1*, which encodes leukocyte-specific protein 1, potentially associated with negative regulation of T cell migration [33], was mostly detected in the *activated* T$_{reg}$ cluster (Figs 3f and S6).

The *naïve* T$_{reg}$ cluster was also characterized by expression of spliced *TCF7* (a marker of T cells with high capacity for self-renewal [34]), *SKAP1* (an immune cell adaptor that regulates T-cell adhesion and optimal cell growth [35]), *RBMS1* (encodes RNA-binding motif 1, a single-stranded-interacting protein involved in helper T cell and T$_{reg}$ post-transcriptional gene regulation [36]), *PTGER2* (encodes PGE2 receptor EP2, involved in differentiation and expansion of helper T cell subsets [37]), and *MALAT1* (a long noncoding RNA linked to regulation of helper T cell differentiation [38]) (Figs 3f, S4 and S6).

In *effector* T$_{reg}$s, splicing-aware differential gene expression analysis performed for the for the *naïve*, *activated* and *effector* T$_{reg}$ clusters identified 58 upregulated spliced genes versus 70 genes revealed by the splicing-unaware approach, with only 14 genes overlapping (threshold log2FC > 1.5, S2 and S1 Tables).

Both approaches indicated upregulation of MHC-II machinery (*HLA-DR/DM/DQ*), consistent with enhanced antigen-specific suppressive capacity of HLA-DR$^+$T$_{reg}$s [39]. Both also highlighted CD39 (*ENTPD1* gene), an ectoenzyme that generates adenosine mediating A2A-dependent immunosuppression [40–42] and supports FOXP3$^+$T$_{reg}$ stability [43], as well as *DUSP4* and *TRIB1*, potentially counter-balancing *effector* T$_{reg}$ activity and proliferation [25,44,45].

Additionally, SANSARA approach captured upregulation of the spliced form of *CD38* (contributes to adenosine-mediated immunosuppression [46], reported as a marker of highly immunosuppressive T$_{reg}$s [47]), *ITGAL* (LFA-1, strong T$_{reg}$-dendritic cells adhesion, critical for T$_{reg}$ homeostasis [48,49]), *LEF1* (FOXP3-cooperating transcription factor stabilizing the T$_{reg}$ program [50]), and *PRF1* (perforin-dependent regulatory functions [51,52]), see Table 1. Together, the spliced gene set was clearly more enriched for T$_{reg}$s effector functionality modules.

## SANSARA investigation of Th1/cytotoxic CD4$^+$ subsets

Next, we focused on analyzing the heterogeneity of gene splicing states in Th1 and cytotoxic CD4$^+$ subsets. In SANSARA analysis, a number of genes characteristic for cytotoxic lymphocytes showed heterogeneous splicing behavior across the clusters, including *NKG7, PRF1, GNLY, GZMA* [57,58], *CCL5, FGFBP2, CST7* [59], *ADGRG1* (*GPR56*) [60], *PLEK* [61], transcription factors *HOPX* [62] and *ETS1* [63] (S7-S9 Figs).

For example, although we detected *PRF1* expression in most Th clusters with conventional splicing-unaware analysis, SANSARA revealed that the spliced form of this gene is almost exclusively expressed in the *Temra cytotoxic Th1* cluster, along with detectable patterns in *Eff-Mem Th1* and *effector* part of T$_{reg}$s (S8 Fig).

Further partitioning of the *Temra cytotoxic Th1* cluster based on splicing of *GNLY* may be indicative of heterogeneous cytotoxic functions performed by distinct subpopulations of helper T cells (S9 Fig).

*CCL5*, which encodes the cytotoxic-lymphocyte–associated chemokine RANTES, was predominantly detected in the unspliced form, except within a compact subpopulation in the *Eff-Mem Th1* cluster—consistent with reports that CCL5

**Table 1. Treg-related spliced genes versus unaware genes upregulated in effector Tregs.**

| Gene/gene set | Spliced gene form (log2FC) | Splicing-unaware (log2FC) | Function in Tregs | Key Refs. |
|---|---|---|---|---|
| MHC-II presentation machinery | Yes (HLA-DRA:2.90; HLA-DMB:2.96; HLA-DQA1:2.37; HLA-DRB1:2.16) | Yes (HLA-DRA:2.20; HLA-DMB:1.70; HLA-DQA1:2.11; HLA-DMA:1.51; HLA-DOA:2.11; HLA-DQA2:2.04; HLA-DRB5:1.75) | HLA-DR+ Tregs are more suppressive. Enhanced antigen-specific suppression. | [39] |
| ENTPD1 (CD39) | Yes (3.03) | Yes (1.53) | Adenosine-mediated immunosuppression. | [40,41] |
| DUSP4 | Yes (4.61) | Yes (1.67) | Phosphatase, potentially counter-balances IL-2/STAT5 signaling. | [25] |
| TRIB1 | Yes (4.59) | Yes (1.98) | Binding partner for FOXP3, TRIB1 overexpression associated with a decrease in Treg proliferative capacity. | [44,45] |
| TSHR | Yes (2.05) | Yes (2.13) | Thyroid-stimulating hormone receptor, increased in Tregs. TSHR inhibition leads to intertumoral Treg depletion. | [53,54] |
| GALNT3 | Yes (2.94) | Yes (1.62) | O-glycosylation, increased expression upon Treg activation. | [55] |
| CD38 | Yes (2.97) | No | Highly immunosuppressive Tregs. Contributes to adenosine-mediated immunosuppression. | [46,47] |
| ITGAL (CD11a/LFA-1) | Yes (9.14) | No | Treg interaction with dendritic cells, homeostasis, homing/retention. | [48,49] |
| LEF1 | Yes (2.91) | No | Transcription factor, cooperates with FOXP3 to stabilize Treg transcriptional program. | [50] |
| LGALS3 | Yes (1.97) | No | Expression increased in human Tregs through a transcriptional mechanism involving ubiquitin D. | [30] |
| PRF1 (perforin) | Yes (2.65) | No | Perforin-dependent regulatory cytotoxicity. | [51,52] |
| ZEB2 | Yes (3.04) | No | Transcription factor that plays a role in TGFβ signaling pathways. | [56] |

is homeostatically produced by memory-phenotype T cells [64], and that its upregulation upon TCR activation proceeds independently of transcription [65] (S7 Fig).

*HOPX*—encoding the transcription factor which is thought to be involved in imprinting for terminal effector differentiation [62,63], was uniformly expressed in *Eff-Mem Th1* and *Temra cytotoxic Th1* clusters in splicing-unaware analysis, but SANSARA revealed distinctive expression patterns for its spliced versus unspliced forms (S8 Fig).

Another transcription factor, ETS1 (which is involved in Th1 differentiation and IFNγ production [63], was uniformly expressed across CD4+ T cells in splicing-unaware analysis. SANSARA showed that spliced *ETS1* is mostly expressed in a compact subpopulation within the *Temra cytotoxic Th1* cluster (S9 Fig).

SANSARA also revealed that a compact subset within the Temra *cytotoxic Th1* cluster is characterized by spliced *ADGRG1*, which encodes a GPR56 protein linked to extracellular signaling and was established as a marker of IFNγ- and TNF-producing Th1 cells [60] (S8 Fig).

## Discussion

The ability to profile single-cell transcriptomes has fundamentally changed our approach to studying the diversity, combinations, and functional impact of genetic programs in living cells [66,67]. However, the functional implementation of genetic programs occurs at multiple levels, not just at the level of the quantity of produced and stored RNA. Ideally, analyzing the transcriptomes of single cells could also reveal the proportion of spliced RNA molecules, which directly affect the functional activity of both mRNAs and non-coding RNAs, as well as offer the insights into alternative splicing [5,68] and trans-splicing [69].

However, this has proven methodologically challenging, as the use of either 5' or 3' end-labeling of RNA molecules with molecular barcodes—alongside inherent limitations of high-throughput sequencing methods—have restricted our ability to comprehensively derive such information for a given RNA molecule [3,68].

Algorithms developed by the Kharchenko and Yosef teams [6,14] have enabled estimation of the RNA processing velocity, making it possible to study transitions between cell types as they differentiate and change gene expression programs at the post-analysis level of scRNA-Seq data. In SANSARA, we have exploited these same algorithms to transform splicing-unaware gene expression data into a splicing-aware format referred to as the saGEX matrix.

SANSARA operates on information about genes predominantly represented in spliced or unspliced form in a given cell, and can be used to build an alternative UMAP data representation that reveals splicing-aware cell clusters. Obtained saGEX matrices are directly usable for Seurat dimensionality reduction and clustering analysis, allowing for seamless transition from conventional scRNA-Seq data analysis. Based on the results obtained, we believe that we managed to find a non-disruptive way to exploit splicing information in scRNA-Seq clustering and dimensionality reduction. Resulting UMAP topology and cluster annotations closely resemble the results of the conventional analysis, and offer an intuitively understandable, and easy-to-implement analytical approach.

The differentiation between spliced and unspliced mRNA enabled by SANSARA facilitates discovery of distinct features that are informative about cell subset heterogeneity. As a demonstration, we have applied SANSARA to peripheral CD4$^+$ T cell scRNA-Seq data, revealing several unexpected features in different helper T cell subsets.

In T$_{reg}$s, we uncovered reciprocal splicing interplay between the master transcription factors FOXP3 and Helios, alongside exclusive expression of the spliced form of *IL10RA* in *activated* and *effector* T$_{reg}$s. Differential expression analysis performed for the *naïve*, *activated* and *effector* T$_{reg}$ clusters using splicing aware and splicing unaware approaches further revealed distinct gene sets enriched in *effector* T$_{reg}$s (S2 and S1 Tables).

Both splicing-aware and splicing-unaware analyses indicated upregulation of the MHC-II presentation machinery - including *HLA-DR*, *HLA-DM*, and *HLA-DQ* genes - required for enhanced antigen-specific suppressive function of effector T$_{reg}$s [39]. Both analyses revealed enrichment of CD39 (*ENTPD1* gene), an ectoenzyme that hydrolyzes extracellular ATP/ADP to AMP, which next mediates downstream immunosuppression via A2A receptors on effector T cells and antigen-presenting cells [40–42], and stabilizes FOXP3$^+$ T$_{reg}$s [43]. Both analyses also demonstrated increased expression of *DUSP4* and *TRIB1*, two genes potentially counterbalancing *effector* T$_{reg}$s functionality. DUSP4 dephosphorylates STAT5 and promotes its turnover, thereby limiting FOXP3-stabilizing STAT5 activity [25]. TRIB1 is a binding partner for FOXP3, which overexpression was associated with a decrease in T$_{reg}$ proliferative capacity [44,45].

Furthermore, only the splicing-aware approach identified several important genes involved in T$_{reg}$ functionality (Table 1). In particular, SANSARA revealed that *effector* T$_{reg}$s express the spliced form of *CD38*, a marker previously associated with highly immunosuppressive T$_{reg}$s [47]. CD38 is an ectoenzyme that uses extracellular NAD$^+$ to produce ADP ribose (ADPR), which can be further converted to AMP [46]. Therefore, CD39 and CD38 contribute complementary ecto-enzymatic cascades that convert extracellular ATP and NAD$^+$ into adenosine, amplifying A2A-mediated immunosuppression when co-expressed on T$_{reg}$s. SANSARA also revealed that *effector* T$_{reg}$s express the spliced form of *ITGAL* (LFA-1, mediates strong T$_{reg}$-dendritic cells adhesion, crucial role in T$_{reg}$ function and homeostasis [48]), *PRF1* (mediates perforin-dependent T$_{reg}$ cytotoxicity [51,52]), and *LEF1* (encodes a transcription factor that cooperates with FOXP3 to stabilize the T$_{reg}$ program [50]).

Altogether, the spliced gene set was more enriched for effector T$_{reg}$ functionality, indicating execution-state mRNA readiness. These findings have significant implications for our understanding of T$_{reg}$ biology [70] and T$_{reg}$-based therapy developments [71], and clearly demonstrate SANSARA's ability to reveal biologically relevant mechanisms that remain hidden to splicing-unaware analyses.

Investigation of Th1 and cytotoxic CD4$^+$ T cells also revealed a number of unexpected splicing-related heterogeneities, indicating a diverse composition of heterogeneous helper T cell functions associated with the type 1 immune response.

Based on these demonstrations, we believe that SANSARA should change the way we analyze single-cell transcriptomic data, opening up a new—and currently unexploited—dimension for investigating the critically important role of splicing regulation in cellular gene expression programs.

## Methods

### saGEX matrix calculation

Raw 5'-RACE 10x Genomics scRNA-Seq data were mapped to the genome using cellranger (v7.1) *count*, taking into account intronic sequences [72]. Importantly, all samples had more than 5,000 median UMI per cell, more than 90,000 reads per cell, and were sequenced with 100 + 100 nt. Subsequently, the velocyto utility [6] was used to count UMIs belonging to unspliced and spliced forms of RNA; cDNAs containing at least some intronic sequences were classified as unspliced, while remaining cDNA reads were identified as spliced. Using the veloVI (v.0.3.0) package [14], we selected highly variable genes (by default – top-2000) and genes with a sufficient number of unspliced and spliced forms for further analysis. The sufficient number was evaluated by the veloVI function `preprocess_data`' by filtering out genes based on linear regression fit and on velocity fit. If velocity 'gamma' coefficient or linear regression coefficient were equal to zero, the gene was discarded from further analysis as poorly detected. For the selected genes, phase portraits reflecting the balance of spliced and unspliced forms were constructed. Out of 2000 highly variable genes, 253, 476 and 476 were selected in three donors for downstream analysis.

The splicing score was calculated for each gene in each cell based on the gene-specific phase portrait using veloVI framework and is basically a velocity value. The normalized expression of variable genes (calculated on conventional cellranger counts via *logNormalise* Seurat function with default parameters) was then multiplied by the splicing score value of each gene in each cell. We divided the resulting metric into spliced and unspliced—negative values were defined as the "expression" of the spliced form of the gene, while positive values described the unspliced form of the gene—and took the modulo values.

The resulting splicing-aware gene expression (saGEX) matrix of spliced/unspliced counts was used for downstream Seurat (v.5.0.1) normalization, dimensional reduction and clustering [73]. All TCR genes were excluded from the variable features used in dimensionality reduction and integration to avoid spurious clusters.

### Integration and clustering

The Harmony pipeline was used for the integration separately for the GEX and saGEX datasets of three donors (3050, 9430 and 9104 cells) [16]. These datasets were independently normalized using the *LogNormalise* function with default parameters and integration features (n = 2997 for splicing-unaware datasets, n = 218 for SANSARA datasets) were selected with the *SelectIntegrationFeatures* function in Seurat. After merging the datasets, variable features of the merged object were set to selected integrated features. Principal Components (PCs) (n = 50) were calculated from scaled integration features. Harmony was run with the default options, and the top 25 corrected Harmony PCs were selected to generate UMAP plots based on the ElbowPlot function in Seurat. Clustering analysis was performed on Harmony PCs via the *FindNeighbors* and *FindClusters* Seurat functions. Under a reasonable number of dimensions (15–30), the results were largely stable. The integrated dataset contained 21584 cells. For comparing clustering between methods, several metrics were used. To compute silhouette scores we used 'cluster' R package (v 2.1.2) at five resolution levels from 0 to 2.5, which corresponds to an increasing number of clusters [74]. Clustering trees were built using the 'clustree' R package for the same set of resolutions from 0 to 2.5 [75]. Trustworthiness and continuity metric were calculated for UMAP/Harmony corrected PCA of SANSARA and conventional analysis by the formulas described in sci-kit learn (sklearn's trustworthiness) and pyDRmetrics python toolkits [76,77]. Jaccard index of similarity and Adjusted Rand Index were calculated by *linkClustersMatrix* and *pairwiseRand* function of 'bluster' R package *(v 1.4.0)* on SANSARA and conventional cluster annotations [78].

## Differential expression and annotation

For differential expression analysis, the *FindMarkers* and *FindAllMarkers* Seurat functions were used. As these data-sets were previously characterized, annotation was performed on the basis of the extensive reference[11], the composition of clusters at resolution level 2.0, and the differential expression results. If cells from the same proposed annotation belonged to several clusters, these clusters were merged. Dotplots and volcano plots were generated via the *DotPlot* Seurat function and '*EnhancedVolcano*' R package [79].

## Supporting information

**S1 Fig. Splicing composition of UMIs chosen for analysis and distributional impact of SANSARA on normalised data. a,b,c**. Relative proportion of spliced versus unspliced UMI per variable gene chosen for downstream analysis (a), per cell (b) and per scRNA-Seq cluster (c) as determined by velocyto. **d,e,f,g**. Comparison of value distribution between conventional expression and SANSARA-generated values. Density plots of log-normalized GEX and saGEX values (d,e). Mean-variance plot of log-normalized GEX and saGEX values as calculated by Seurat. Dots correspond to genes (f,g).
(TIF)

**S2 Fig. Integration, cluster stability. a,b.** Harmony integration of scRNA-Seq data for the three donors performed with conventional (a) and splicing-aware (b) datasets. **c.** Clustering at different UMAP resolutions. **d,e.** Clustering trees for splicing-unaware (d) and splicing-aware (e) datasets.
(TIF)

**S3 Fig. Quantifying clustering between splicing-unaware and SANSARA methods. a.** Comparison of silhouette scores on multiple resolutions between conventional splicing-unaware and SANSARA methods. Larger score points to greater separation of the clusters. **b**. Trustworthiness and continuity metrics at three k-values (5, 15, 30) for splicing-unaware and SANSARA dimensionality reduction step from PCA to UMAP. Values reflect the relative preservation of neighbors in UMAP compared to PCA. **c**. Left: Correspondence between clusters produced by splicing-unaware analysis and SANSARA. Each row identifies the cross-mapping of clusters from the different methods, normalized by the cluster abundance as calculated by Jaccard index of similarity. Right: Adjusted Rand Index. Pairwise heatmap shows which clusters of the reference (conventional splicing-unaware analysis) retain their integrity in SANSARA clustering. Higher index means the two clustering algorithms agree on which cells belong together and which are separated.
(TIF)

**S4 Fig. Selected genes characterized by heterogeneous expression of spliced and unspliced forms.** Splicing-unaware UMAP plots are shown at left; center and right panels show splicing-aware UMAP plots. *FOS*—encoding a c-Fos protein which interacts with c-Jun, forming heterodimeric AP-1 transcription factor that prominently affects CD4+ T cell differentiation [80]. *ANXA1*—encoding Annexin A1, the key driver of glucocorticoid anti-inflammatory effects, involved in T-cell differentiation, altering the strength of TCR signaling [81] and Th1-Th2 counterbalance driven by GATA3 and TBX21 transcription factors [82]. *TCF7*—encoding transcription factor T cell factor 1 which marks CD4 + T cells ability to self-renew [34] and which expression goes down along with effector T cell differentiation [83], especially towards CD4 + cytotoxic T cells [84]. *INPP4B*—encoding inositol poly-phosphate 4-phosphatase that was suggested to play role in T cell proliferation, survival and differentiation [85]. *MALAT1*—long noncoding RNA, reported as regulator of helper T cell differentiation from naïve CD4 + T cells [38]. *ACTG1* and *ACTB*—cytoskeleton-related protein genes [32].
(TIF)

**S5 Fig. Selected genes characterized by heterogeneous expression of spliced and unspliced forms.** Violin plots of splicing-unaware (top) and -aware (middle, bottom) gene expression across clusters are shown. *FOS*—encoding a

c-Fos protein which interacts with c-Jun, forming heterodimeric AP-1 transcription factor that prominently affects CD4$^+$ T cell differentiation [80]. *ANXA1*—encoding Annexin A1, the key driver of glucocorticoid anti-inflammatory effects, involved in T-cell differentiation, altering the strength of TCR signaling [81] and Th1-Th2 counterbalance driven by GATA3 and TBX21 transcription factors [82]. *TCF7*—encoding transcription factor T cell factor 1 which marks CD4 + T cells ability to self-renew [34] and which expression goes down along with effector T cell differentiation [83], especially towards CD4 + cytotoxic T cells [84]. *INPP4B*—encoding inositol poly-phosphate 4-phosphatase that was suggested to play role in T cell proliferation, survival and differentiation [85]. *MALAT1*—long noncoding RNA, reported as regulator of helper T cell differentiation from naïve CD4 + T cells [38]. *ACTG1* and *ACTB*—cytoskeleton-related protein genes [32].
(TIF)

**S6 Fig. Selected genes characterized by heterogeneous expression of spliced and unspliced forms in T$_{reg}$ clusters.** The lefthand column shows splicing-unaware UMAP plots, center and righthand columns show splicing-aware UMAP plots.
(TIF)

**S7 Fig. Heterogeneous expression of spliced and unspliced forms of *CCL5, GZMA, NKG7,* and *CST7*.** The lefthand column shows splicing-unaware UMAP plots for comparison.
(EPS)

**S8 Fig. Heterogeneous expression of spliced and unspliced forms of *HOPX, PRF1, ADGRG1,* and *LYAR*.** The lefthand column shows splicing-unaware UMAP plots for comparison.
(EPS)

**S9 Fig. Heterogeneous expression of spliced and unspliced forms of *ETS1, CLIC1, GNLY,* and *FGFBP2*.** The lefthand column shows splicing-unaware UMAP plots for comparison.
(EPS)

**S1 Table. Splicing-aware and splicing-unaware differential gene expression analysis across helper T cell scRNA-Seq clusters.**
(XLSX)

**S2 Table. Splicing-aware and splicing-unaware differential gene expression analysis across *naïve*, *activated*, and *effector* T$_{reg}$ scRNA-Seq clusters.**
(XLSX)

## Author contributions

**Conceptualization:** Dmitry M. Chudakov.

**Data curation:** Daniil K Lukyanov, Evgeniy S Egorov, Denis Syrko, Victor V Kotliar.

**Investigation:** Daniil K Lukyanov, Valeria V Kriukova, Kristin Ladell, David A. Price, Andre Franke.

**Methodology:** Daniil K Lukyanov, Evgeniy S Egorov, Dmitry M. Chudakov.

**Project administration:** Dmitry M. Chudakov.

**Resources:** Denis Syrko, Victor V Kotliar.

**Software:** Evgeniy S Egorov.

**Supervision:** Dmitry M. Chudakov.

**Visualization:** Dmitry M. Chudakov.

**Writing – original draft:** Daniil K Lukyanov, Dmitry M. Chudakov.

**Writing – review & editing:** Dmitry M. Chudakov.

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
