## [Decision Letter · Decision Letter 0]

18 Jun 2025

Splicing-aware scRNA-Seq resolution

PLOS Computational Biology

Dear Dr. Chudakov,

Thank you for submitting your manuscript to PLOS Computational Biology. After careful consideration, we feel that it has merit but does not fully meet PLOS Computational Biology's publication criteria as it currently stands. Therefore, we invite you to submit a revised version of the manuscript that addresses the points raised during the review process. In addition to the comments from Reviewers 1 and 2, please address the following comments from the reviewer 3:

In their paper, the authors describe SANSARA (Splicing-Aware scrNa-Seq AppRoAch), a method that generates a splicing-adjusted gene expression matrix (saGEX) that estimates the amount of splicing for each gene in each cell. The resulting saGEX is then subjected to a conventional clustering and dimensionality reduction pipeline to reconstruct a representation of splicing that leads to the scRNA-Seq data. This is a very promising approach that can be used by many immunologists in the field.

They applied this technique to characterise the splicing of human peripheral blood helper T cells, specifically regulatory T cell (Treg) subsets. This has revealed the crosstalk between the master transcription factors FoxP3 and Helios, Galectin3 - which plays a major role in regulating the cell fate of regulatory cells - and many other important features of the splisosome of Tregs. However, there are several points that require further attention. In particular, a more detailed description at several positions would really help to understand the study in more detail.

At this stage, I have several major comments:

1) In Figure 3, the authors take the top three hits, FOXP3, Helios and IL-10R. They then go on to describe hit number 6, which is galectin. In my opinion, it would be important to also show the UMAP profiles for DUSP4 and CARD16, which are also very important regulators of T cells, and to devote some attention to them in the manuscript.

2) The discussion is very short and should be expanded. The authors pretty much repeat what they wrote in their abstract: "… in Tregs, we uncovered reciprocal splicing interplay between the master transcription factors FoxP3 and Helios, alongside exclusive expression of the spliced form of IL10RA in activated and effector Tregs. These findings have significant implications for our understanding of Treg biology and Treg-based therapy developments. .. However, it would be important to expand which exactly implications it might have. Furthermore, they do not mention the other hits in the discussion like galectin3, which would be benefitial.  

3) Materials and methods: the description is rather short and all chapters need to be expanded with more concrete information:

For example, the authors write:

Using the veloVI (v.0.3.0) package10, we selected highly variable

genes and genes with a sufficient number of unspliced and spliced forms for further

analysis. ..

The Questions: How many genes were selected, what was the threshold value for the number of the spliced forms?

…. The normalized expression of variable genes was then multiplied by the splicing score value of each gene in each cell..

The question is  how the splicing score was calculated?

Please submit your revised manuscript within 60 days Aug 18 2025 11:59PM. If you will need more time than this to complete your revisions, please reply to this message or contact the journal office at ploscompbiol@plos.org. Please include the following items when submitting your revised manuscript:

We look forward to receiving your revised manuscript.

Kind regards,

Inna Lavrik

Academic Editor

PLOS Computational Biology

James R. Faeder

Section Editor

PLOS Computational Biology

**Journal Requirements:**

1) Please provide an Author Summary. This should appear in your manuscript between the Abstract (if applicable) and the Introduction, and should be 150-200 words long. The aim should be to make your findings accessible to a wide audience that includes both scientists and non-scientists. Sample summaries can be found on our website under Submission Guidelines:

4) We notice that your supplementary Figures are included in the manuscript file. Please remove them and upload them with the file type 'Supporting Information'. Please ensure that each Supporting Information file has a legend listed in the manuscript after the references list.

**Reviewers' comments:**

Reviewer's Responses to Questions

**Comments to the Authors:**

Reviewer #1: See the attached file.

**Have the authors made all data and (if applicable) computational code underlying the findings in their manuscript fully available?**

Reviewer #1: Yes

PLOS authors have the option to publish the peer review history of their article (what does this mean? ). If published, this will include your full peer review and any attached files.

**Do you want your identity to be public for this peer review?** For information about this choice, including consent withdrawal, please see our Privacy Policy .

Reviewer #1: No

**Figure resubmission:**

**Reproducibility:**



---

## [Editor Report · Decision Letter 1]

28 Oct 2025

Dear Dr. Chudakov,

We are pleased to inform you that your manuscript 'Splicing-aware scRNA-Seq resolution reveals execution-ready programs in effector Tregs' has been provisionally accepted for publication in PLOS Computational Biology.

Best regards,

Inna Lavrik

Academic Editor

PLOS Computational Biology

James Faeder

Section Editor

PLOS Computational Biology

---

## [Editor Report · Acceptance letter]

PCOMPBIOL-D-24-02030R1

Splicing-aware scRNA-Seq resolution reveals execution-ready programs in effector Tregs

Dear Dr Chudakov,

I am pleased to inform you that your manuscript has been formally accepted for publication in PLOS Computational Biology. Your manuscript is now with our production department and you will be notified of the publication date in due course.

With kind regards,

Anita Estes
